# Seroprevalence and risk factors of hantavirus and hepatitis E virus exposure among wildlife farmers in Vietnam

Ha Thi Thanh Nguyen[1,2,*], Hu Suk Lee[2,3], Bernard Bett[4], Jiaxin Ling[1],
Thang Nguyen-Tien[5], Sinh Dang-Xuan[2], Hung Nguyen-Viet[4], Fred Unger[2],
Steven Lâm[4], Vuong Nghia Bui[6], Tung Duy Dao[6], Åke Lundkvist[1],
Genevieve Cattell[1,2], Johanna F. Lindahl[1,2,7,*]

1 Zoonosis Science Center, Department of Medical Biochemistry and Microbiology, Uppsala University, Uppsala, Sweden, 2 International Livestock Research Institute, Hanoi, Vietnam, 3 College of Veterinary Medicine, Chungnam National University, Daejeon, Republic of Korea, 4 International Livestock Research Institute, Nairobi, Kenya, 5 University of Texas Medical Branch, Galveston, Texas, United States of America, 6 National Institute of Veterinary Research, Hanoi, Vietnam, 7 Swedish Veterinary Agency, Uppsala, Sweden

* ha.nguyen@imbim.uu.se or h.t.nguyen@cgiar.org (HTTN); johanna.lindahl@imbim.uu.se or johanna.lindahl@sva.se (JFL)

## Abstract

### Background

Wildlife farming is a growing industry, but it poses substantial risks for zoonotic disease transmission, including infections caused by hantaviruses and hepatitis E virus (HEV). This study aimed to determine seroprevalences of these viruses among wildlife farmers and identify associated risk factors.

### Methods

A cross-sectional study was conducted among 210 wildlife farmers in Lao Cai and Dong Nai provinces in Vietnam who raised bats, bamboo rats, civets, and wild boars. Of these, 207 provided serum samples for serological testing for hantavirus and HEV antibodies. Apparent (AP) and true (TP) prevalences were estimated, and multivariable logistic regression was performed to identify risk factors.

### Results

The AP of hantavirus IgG was 8.7%, 95% confidence interval (CI): 5.4–13.6 (TP: 4.7%, 95% credible interval (CrI): 0.2–11.1). HEV IgG AP was 26.7%, 95%CI: 20.8–33.2 (TP: 27.1%, 95%CrI: 21.3–33.4). Hantavirus IgM testing was also performed due to higher IgG seroprevalence compared to earlier studies, detecting IgM antibodies in 1.9% of samples (95%CI: 0.6–5.2) (TP: 1.7%, 95%CrI: 0.1–4.7). Hantavirus seropositivity was significantly associated with engaging only in wildlife farming, and not participating in other activities such as hunting, trading, slaughtering, processing,

**Data availability statement:** All relevant data are within the manuscript and its Supporting information files.

**Funding:** This work was initiated as part of the CGIAR initiative on One Health and subsequently finalised under the Sustainable Animal and Aquatic Foods Program, both of which are supported by contributors to the CGIAR Trust Fund (https://www.cgiar.org/funders). In addition, JFL's time was funded by FORMAS (2021–00833), HSL's time was supported by the National Research Foundation of Korea (NRF) grant funded by the Korea government (MSIT) (RS-2022-NR068754). The funders had no role in study design, data collection and analysis, decision to publish, or preparation of the manuscript.

**Competing interests:** The authors have declared that no competing interests exist.

guano collection, or consumption (OR = 2.7, 95% CI: 1.1–6.9). HEV seropositivity was significantly associated with men gender (OR = 3.1, 95%CI: 1.4–7.3), older age (OR = 1.03, 95%CI: 1.0–1.1), raw meat consumption (OR = 6.8, 95%CI: 1.6–31.8), residing at higher altitudes (OR = 31.6, 95%CI: 5.5–204.4), and reporting use of protective clothing (OR = 4.0, 95%CI: 1.4–11.2), although their proper use was not assessed.

## Conclusions

This study highlights behavioural and environmental risk factors associated with wildlife farming and zoonotic pathogens exposure. Public health interventions should focus on biosecurity, proper hygiene practices, and risk communication to reduce the transmission in wildlife farming settings.

## Introduction

"Wildlife farming" refers to the practice of breeding or keeping wild animals in captivity in order to later sell the animals or their products for different purposes, including food, fertilizer, traditional medicine, clothing, decorations, experiments, and pets [1–6]. While wildlife farming is actively encouraged in many countries, with 90 countries reporting to have commercial wildlife breeding operations [1], as a way to capitalise on its economic value and potentially alleviate pressure on wild animal populations, several studies suggested this practice may have important consequences [2,7,8]. Many wildlife species commonly farmed and traded, such as bats [9–13], rodents [14–18], and wild boars [19,20] are natural reservoirs of zoonotic pathogens [21], increasing the potential for disease spillover to humans.

Among emerging zoonotic diseases, hantavirus and hepatitis E virus (HEV) pose substantial challenges to policymakers and public health systems due to their widespread nature and rapid emergence. Hantaviruses are single-stranded, tri-segmented, negative-sense RNA viruses, belonging to the *Hantaviridae* family, primarily carried by rodents but also found in bats, shrews, and moles [22]. Human infection occurs through contact with contaminated urine, saliva and faeces, leading to haemorrhagic fever with renal syndrome (HFRS) or hantavirus pulmonary syndrome (HPS), both of which have high morbidity and mortality rates [23]. Globally, hantaviruses are categorised into "Old World" species, which cause HFRS, including Hantaan virus (HTNV), Seoul virus (SEOV), Puumala virus (PUUV), and Dobrava-Belgrade virus (DOBV). And the "New World" hantaviruses, including Sin Nombre virus (SNV), New York virus, Black Creek Canal virus, and Andes virus (ANDV), are the cause of HPS [24]. Notably, ANDV is the only hantavirus confirmed for person-to-person transmission [25]. In Vietnam, previous studies have reported SEOV and HTNV in wild rodents and evidence of human exposure [26–30]. HEV is also a single-stranded, positive-sense RNA virus, belonging to the *Hepeviridae* family, that are transmitted through oral consumption of contaminated food and water or through direct contact with infected animals [31]. HEV is a major public health concern, with

an estimated 20 million infections occurring globally each year, accounting for over three million symptomatic cases and approximately 56,600 deaths worldwide [32,33]. HEV is divided into four main genotypes that cause human diseases (genotypes 1, 2, 3, and 4) with distinct host ranges and transmission pathways [32–34]. Genotypes 1 and 2 infect only humans via faecal-oral route, while genotypes 3 and 4 are transmitted through contact with infected animals or consumption of contaminated food or water [31–34]. Previous studies have isolated genotypes 3 and 4 in humans and various species of wild animals, including wild boars, rodents, and deer, which are all commonly farmed, indicating zoonotic transmission [20,35–40]. Additionally, rat HEV, a novel HEV strain that causes human hepatitis, has been detected in rats [41], and serological evidence of its infection has been observed in humans in Vietnam [42]. Suspected routes of rat HEV include contact with animal excreta, contaminated food or water, and environmental exposure [43].

Both hantavirus and HEV infections may cause severe consequences, with hantavirus mortality rates ranging from <1% to 15% [44], and HEV mortality rates are notably high among pregnant women infected with genotypes 1 or 2, while HEV genotype 3 and 4 infections can range in severity from asymptomatic to fulminant hepatitis [45,46]. Similarly, rat HEV infections in humans can range from mild, self-resolving illnesses to life-threatening acute liver failure and chronic infections in individuals with weakened immune systems [47]. These findings highlight the relevance of investigating wildlife farming contexts, where contact with potential animal reservoirs and consumption of wild meat may facilitate zoonotic hantavirus and HEV exposure.

Vietnam has extensive captive wildlife farming and trade with at least 6,744 captive wildlife farms located in 54 out of 63 provinces in the country, raising species known to carry zoonotic pathogens [48]. Previous studies have demonstrated human exposure to hantavirus (prevalence ranged 1.6% – 3.7% [30,49,50]) and HEV (prevalence ranged 16% – 72.1% [51,52]). Research in both endemic and non-endemic areas has revealed demographic and behavioural risk factors associated with hantavirus and HEV infections. Hantavirus risk is linked to exposure to rodent excreta [53], while HEV is associated with older age, man gender, poor sanitation, undercooked meat, and animal contact in farming settings [32,33,54–56]. However, limited research in Vietnam has explored hantavirus and HEV prevalence among wildlife farmers, particularly in settings with frequent wildlife-human interactions [21], and their behaviours and other factors associated with increased risk of zoonotic disease transmission.

Serology is an important approach for detecting past or present exposures to hantavirus and HEV, even in asymptomatic individuals. By identifying seroprevalence and risk factors that influenced the exposure, serological studies can support early detection and prevention of zoonotic spillovers, crucial for One Health approaches, which emphasize the interconnectedness of human, animal, and environmental health [57]. In this context, improved understanding can help develop targeted surveillance and inform public health strategies to reduce the risk of transmission of zoonotic diseases from farmed wild animals. In Southeast Asia, few studies have been conducted to evaluate the seroprevalence of these pathogens in humans, including ruminant farmworkers [58], occupationally exposed individuals (slaughterhouse staff, pig farmers, pork vendors) [54], and people living in farming communities [50], however, specific data on hantavirus and HEV seroprevalence among wildlife farmers are lacking.

Drawing on a case study of Vietnam, this study investigated the seroprevalence of hantavirus and HEV antibodies among wildlife farmers. By examining demographic, occupational, environmental, and behaviour risk factors of wildlife farmers in Lao Cai and Dong Nai provinces, this research aims to provide critical insights into the epidemiology of zoonotic diseases in this high-risk occupational group, contributing to the broader understanding of zoonotic disease transmission from wildlife pathogens in Vietnam with implications for Southeast Asia.

## Methods

### Ethics statement

Written informed consent was obtained from all participants before the interview and sample collection. For participants unable to read or write, the consent form was read aloud, and explanations were provided, then informed consent was

obtained through thumbprints from participants who were unable to sign. To ensure participant anonymity, unique identifiers were assigned to each participant for sample identification and to facilitate the sharing of results from the tests. The study was approved by the Ethical Review Committee of Hanoi University of Public Health (approval number 338/2023/YTCC-HD3).

## Study design

A cross-sectional study design was employed to collect samples and data from wildlife farmers in Lao Cai and Dong Nai provinces in Vietnam between August 2023 and March 2024.

## Study location

The study focused on two provinces with distinct characteristics related to wildlife trade: Lao Cai in the north and Dong Nai in the south (Fig 1). Lao Cai and Dong Nai represent two ecologically and economically distinct regions with different patterns of wildlife farming and trade. Lao Cai, a mountainous province bordering China, is characterized by small-scale wildlife farming integrated with hunting, home processing, and informal trade. The province has a history of efforts aiming to address this issue, including the "Saving Species" project [59], which brought together diverse stakeholders such as the Provincial People Committee, the Department of Forestry Protection (DFP), and the Department of Animal Health. Dong

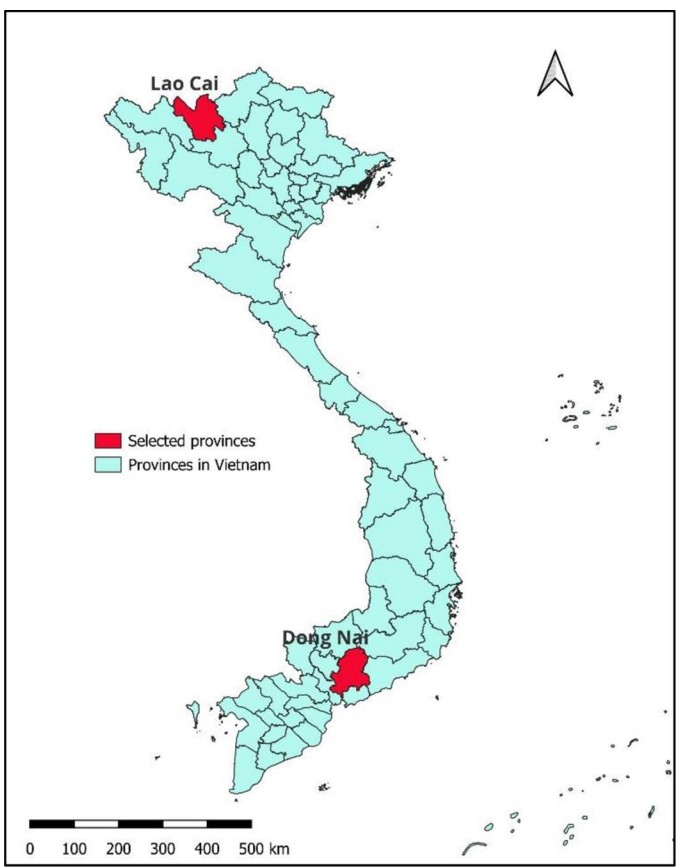

**Fig 1. Map of Vietnam and locations of the two provinces selected for the study.** Source of Vietnam administrative boundary data: https://data.humdata.org/dataset/cod-ab-vnm, accessed March 01, 2025, licensed under CC-BY-IGO.

Nai, in contrast, is a lowland province in southern Vietnam with a high concentration of commercial wildlife farms. Farmers in both provinces may be exposed through contact with farmed wildlife, but in Lao Cai, additional risks may arise from hunting and butchering, while in Dong Nai, risks may extend along the commercial chain. This contrasting selection of provinces allowed a comparative analysis of wildlife trade dynamics within Vietnam.

The study focused on the farms with four species: bats, bamboo rats, civets, and wild boars. The selected farms varied in size and purpose, including small-scale household operations, medium-sized, and larger-scale commercial farms. While some farms focused solely on wildlife farming, others were also involved in trading, slaughtering, processing, harvesting bat guano, or consumption. To investigate wildlife farms and associated risks, six districts in Lao Cai (Lao Cai city, Bao Thang, Bao Yen, Van Ban, Bat Xat, and Muong Khuong) and six districts in Dong Nai (Bien Hoa city, Dinh Quan, Tan Phu, Vinh Cuu, Trang Bom, and Thong Nhat) were selected. These districts were chosen since they had a higher number of wildlife farms reported by the DFP as of December 31, 2022. We prioritized districts with the highest concentration of bamboo rat and civet farms. Given the DFP's limited data on bat caves and wild boar/indigenous pig farms, these were selected based on accessibility and proximity to potential infection sources. Specifically, we identified bat caves and selected nearby wild boar farms for inclusion in the study. Due to the smaller number of wildlife farms in Lao Cai (72 farms in total), all 46 farms within the six selected districts were included. To compensate for the limited sample size in Lao Cai, the rest of the farms were randomly recruited from Dong Nai. In Dong Nai, we recruited 164 wildlife farms from six selected districts, accounting for 24.5% (164/ 669) of the province's wildlife farms. The map of the study sites was developed using QGIS version 3.30.0.

## Target population and sample size estimation

All study participants were adult wildlife farm owners or workers aged 18 or older who provided informed consent. The participants were required to have at least six months of continuous involvement in wildlife farming to ensure sufficient exposure time for potential hantavirus or HEV infection and subsequent antibody development. Individuals not directly involved in the farming of wild animals, or with a known history of confirmed hantavirus or HEV infection, or receiving immunosuppressive therapy, were excluded. Eligible individuals who declined participation were replaced by other consenting participants. We recruited one participant per farm, typically selecting either the owner or the individual most actively engaged in daily farming tasks.

A sample size calculation was conducted to determine the minimum number of participants required for the study, assuming *priori* seroprevalences of both hantavirus and HEV of 15% [60,61] and that the study would estimate the seroprevalences of these pathogens with a 5% margin of error at a 95% confidence level (p = 0.15, z = 1.96, ε = 0.05). These assumptions provided a sample size of 196 (see the formula below).

$$n = \frac{z^2 \times p\,(1-p)}{\varepsilon^2}$$

To account for potential sample losses, an additional 10% buffer was added to the target sample size, resulting in a total of 216 potential participants. In total, 210 people were recruited in the study.

## Collection of serum samples and socio-demographic data

Five millilitres (5 mL) of venous blood samples were collected via venipuncture from the median cubital vein, which was performed by a qualified clinician from the provincial centre for disease control (CDC). The venipuncture site was cleaned with alcohol swabs and allowed to air dry for 30 seconds. All blood samples were dispensed into labelled clot activation tubes immediately after collection. These tubes were incubated at room temperature to ensure complete coagulation. Subsequently, the sera were separated, aliquoted into labelled cryovials, maintained on ice, and transported to the National Institute of Hygiene and Epidemiology for serological analysis.

The questionnaire (Supporting information S1 File) designed to collect socio-demographic and relevant epidemiological data was developed in KoboToolbox (https://www.kobotoolbox.org/) and pre-tested to ascertain comprehensibility. Minor revisions were made afterward based on the feedback from pre-testing sessions before final dissemination [62]. The face-to-face interview took place in the wildlife farmers' households and lasted for around 45 minutes. The data and sample collection in the study is presented in Fig 2.

## Screening of serum for anti-Hantavirus and anti-HEV antibodies

Hantavirus infections were confirmed by detecting hantavirus-specific IgG and IgM antibodies in participants' serum using Focus DxSelect™ ELISA kits (Focus Diagnostics, Cypress, CA, USA). The enzyme-linked immunosorbent assay (ELISA) kits utilized a recombinant nucleoprotein cocktail to detect antibodies against a broad range of hantavirus strains, including SEOV, HTNV, PUUV, DOBV, and SNV. To investigate recent hantavirus infection, all specimens were further tested for anti-hantavirus IgM. Sensitivity and specificity vary depending on the specific hantavirus strains. According to the material supplied by the manufacturer, the IgM test has an overall sensitivity of 95.1% (95% confidence interval (CI): 83.5–99.4%) and a specificity of 94.1% (95%CI: 83.8–98.8%). The IgG test has comparable performance characteristics, with an overall sensitivity of 95% (95%CI: 75–98%) and an overall specificity of 95% (95%CI: 91.4–100%).

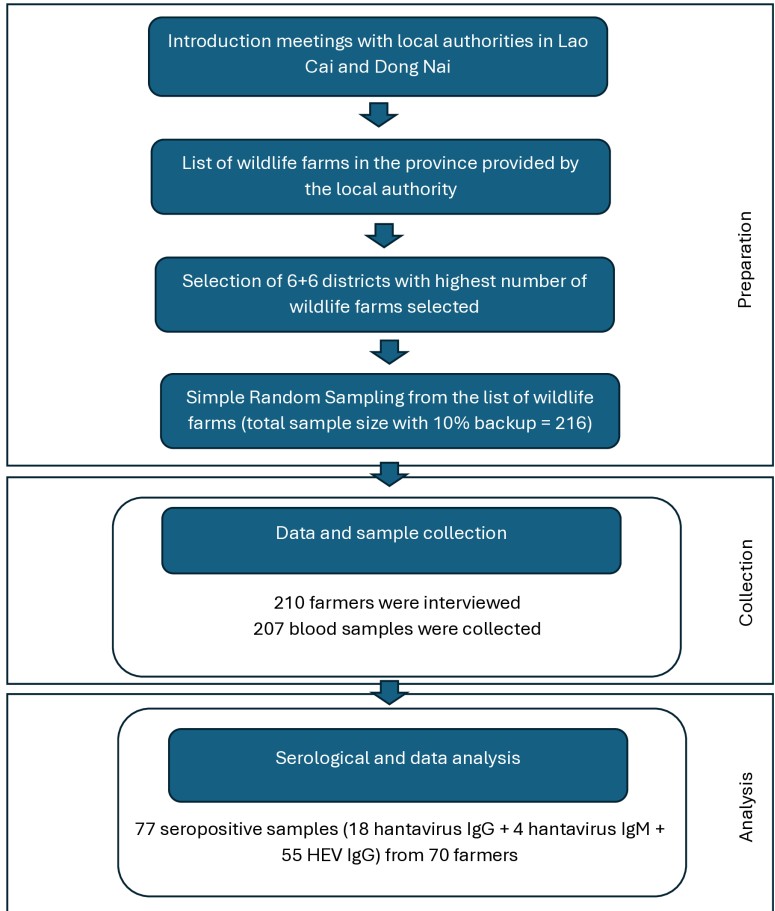

**Fig 2. Workflow of data and sample collection and analysis of hantavirus and HEV in Vietnam.**

Serum samples from wildlife farmers were also screened for anti-HEV IgG using the Wantai HEV IgG ELISA kit (Beijing Wantai Biological Pharmacy Enterprise, Beijing, China). The Wantai HEV IgG assay targets the open reading frame 2 (ORF2) capsid protein, allowing detection of prior HEV exposure across all genotypes [63]. This assay is recognized for its high sensitivity in detecting HEV antibodies [64,65], making it a valuable tool and considered a "gold standard" for HEV seroprevalence studies [66]. The test is currently one of the most commonly used assays with reported sensitivity and specificity of the HEV IgG of 99.1% (95%CI: 96.4–99.8%) and 99.9% (95%CI: 96.8–99.9%), respectively.

Testing and calculations (specifically, sample-to-cut-off ratio, optical density, and determination of positive, equivocal, and negative results) for both anti-Hantavirus and anti-HEV antibodies were performed following the manufacturer's instructions. In addition, for each round of ELISA assays, we included three negative and two positive controls to validate the results. By combining IgG and IgM results, we aimed to maximize the detection of Hantavirus exposure across varying stages of infection, and a participant was classed as positive if they had a positive test result in either one of the tests.

## Data analysis

For the quantitative analyses, data from KoboToolbox were transferred to R version 4.4.2 for management and analysis. Seroprevalence was calculated by dividing the number of testing positive by the number of tested and 95% confidence intervals were determined using the proportion test in R.

$$\text{Seroprevalence} = \frac{\text{Number of positive}}{\text{Number of tested}} \times 100$$

The true prevalence (TP) of anti-hantavirus and anti-HEV antibodies in serum samples were estimated following the methodology of Lee et al. [67]. The TP represents the actual proportion of individuals with antibodies against these viruses within the population, while apparent prevalence (AP) is based on diagnostic test results. The TP was derived from AP by incorporating the test sensitivity and specificity of the diagnostic kits using a Bayesian analysis [68]. Prior distributions for sensitivity and specificity were based on ELISA experiments, with a non-informative Jeffreys prior (beta = 0.5, 0.5) assigned to TP [48]. Markov Chain Monte Carlo (MCMC) sampling was performed in R using JAGS via the 'rjags' package in R [69,70], discarding the first 1000 iterations as burn-in, and using 10,000 iterations for posterior inference. The AP was calculated with a 95% Clopper-Pearson/Exact confidence interval and TP was estimated as the posterior median with a 95% credible interval (CrI). Differences between prevalence estimates were assessed by credible interval overlap. The MCMC convergence was assessed using trace plots, posterior density, Brooks-Gelman-Rubin (BGR) plots, and autocorrelation plots via the CODA package [71].

Univariable analyses were performed using the generalized linear model function 'glm' in R to identify potential risk factors associated with seropositivity. Variables with p-values ≤ 0.05 were considered statistically significant and included in the multivariable logistic regression model. Multivariable models were fitted using function 'glmer' in R with the province as a random effect. In the initial model, we included both statistically significant variables with p-values ≤ 0.05 and biologically plausible variables with p-values > 0.05 to minimize the risk of missing important factors. We then applied a backward stepwise regression approach to identify the best-fitting models. Missing values were omitted before the regression analysis process. Model fit was assessed using the Hosmer-Lemeshow goodness-of-fit test implemented in the 'generalhoslem' R package. In addition to demographic and behavioural variables, we incorporated mammalian diversity as an environmental factor into our analysis for a more comprehensive understanding of the factors influencing seropositivity. Mammalian diversity data was downloaded from https://www.iucnredlist.org/resources/other-spatial-downloads#SR_2024, reflecting mammal species' richness and diversity within the study region. Factors associated with any seropositivity, anti-hantavirus and anti-HEV were determined based on p-values ≤ 0.05, odds ratios (ORs), and 95% confidence interval (CI).

## Results

### Socio-demographic characteristics of wildlife farmers

A total of 210 wildlife farmers completed a quantitative survey, the socio-demographic characteristics of the participants are shown in Table 1.

Most participants were from Dong Nai (78.1%), with 21.9% from Lao Cai, and included 62.4% men and 37.6% women. Participants' ages ranged from 22 to 77 years, with a median age of 45, and there was no significant difference between

**Table 1. Socio-demographic characteristics of the participants from Lao Cai and Dong Nai.**

|  | Lao Cai (n, %) | Dong Nai (n, %) | Overall (n, %) |
|---|---|---|---|
| **Participants** | 46 (21.9%) | 164 (78.1%) | 210 (100%) |
| **Gender** | | | |
| Man | 30 (65.2%) | 101 (61.6%) | 131 (62.4%) |
| Woman | 16 (34.8%) | 63 (38.4%) | 79 (37.6%) |
| **Age** (median, min-max) | 46 (22–77) | 45 (23–70) | 45 (22–77) |
| **Ethnicity** | | | |
| Kinh | 20 (43.5%) | 151 (92.1%) | 171 (81.4%) |
| Ethnic minority | 26 (56.5%) | 13 (7.9%) | 39 (18.6%) |
| **Education** | | | |
| No formal education | 3 (6.5%) | 4 (2.4%) | 7 (3.3%) |
| Primary school | 6 (13.1%) | 20 (12.2%) | 26 (12.4%) |
| Secondary school | 18 (39.1%) | 64 (39.0%) | 82 (39.1%) |
| High school | 12 (26.1%) | 50 (30.5%) | 62 (29.5%) |
| College or higher level | 7 (15.2%) | 26 (15.9%) | 33 (15.7%) |
| **Additional occupation** | | | |
| Trading/ self-employed | 8 (17.4%) | 47 (28.7%) | 55 (26.2%) |
| No other occupation | 7 (15.2%) | 45 (27.4%) | 52 (24.8%) |
| Plant farming/crop cultivation | 17 (37.0%) | 34 (20.7%) | 51 (24.3%) |
| Livestock and poultry farming | 9 (19.6%) | 15 (9.2%) | 24 (11.4%) |
| Government employee | 2 (4.3%) | 14 (8.5%) | 16 (7.6%) |
| Private company | 3 (6.5%) | 9 (5.5%) | 12 (5.7%) |
| **Species farmed** | | | |
| Civets | 20 (43.5%) | 68 (41.5%) | 88 (41.9%) |
| Wild boars | 15 (32.6%) | 39 (23.8%) | 54 (25.7%) |
| Bamboo rats | 8 (17.4%) | 44 (26.8%) | 52 (24.8%) |
| Bats | 3 (6.5%) | 13 (7.9%) | 16 (7.6%) |
| **Specific wildlife activities** | | | |
| Consuming wild meat | 32 (69.6%) | 99 (60.4%) | 131 (62.4%) |
| Trading live wild animals | 28 (60.9%) | 63 (38.4%) | 91 (43.3%) |
| Processing | 23 (50%) | 43 (26.2%) | 66 (31.4%) |
| Slaughtering | 20 (43.5%) | 33 (20.1%) | 53 (25.2%) |
| Farming wild animals only | 4 (8.7%) | 45 (27.4%) | 49 (23.3%) |
| Trading slaughtered wild animals | 7 (15.2%) | 13 (7.9%) | 20 (9.5%) |
| Hunting/trapping | 9 (19.6%) | 6 (3.7%) | 15 (7.1%) |
| Harvesting bat guano | 0 (0.0%) | 14 (8.5%) | 14 (6.7%) |
| Consuming other wildlife products | 3 (6.5%) | 7 (4.3%) | 10 (4.8%) |

the two provinces. Educational levels varied, with the majority having attended secondary school (39.0%), followed by high school (29.5%). Among those, 15.7% had a college or higher education, and 3.3% had no formal education.

Of these, 24.8% of participants reported no occupation other than wildlife farming, while others engaged in trading/self-employment (26.2%) and plant farming (24.3%). Regarding engagement with wildlife, civets were the most commonly farmed species in both provinces, with 43.5% of participants in Lao Cai and 41.5% in Dong Nai reporting civet farming. Wild boars were the second most commonly farmed species, with 32.6% in Lao Cai and 23.8% in Dong Nai. Then followed by bamboo rats, particularly in Dong Nai (26.8%) compared to Lao Cai (17.4%).

Regarding specific wildlife activities, many wildlife farmers not only farmed wild animals but also held many other roles, such as hunters or trappers, traders, butchers, and consumers. Common activities included consuming wild meat (62.4%) and trading live wild animals (43.3%). Participants in Lao Cai were more actively engaged in hunting/trapping (19.6%) compared to Dong Nai (3.7%). Similarly, a greater proportion of participants in Lao Cai reported involvement in slaughtering (43.5%) and processing wildlife (50.0%) compared to Dong Nai (20.1% and 26.2%, respectively). Trading live wild animals was also more common in Lao Cai (60.9%) than Dong Nai (38.4%). On the other hand, harvesting bat guano was exclusively reported in Dong Nai (8.5%), and wildlife farming-only activities were also more prominent in Dong Nai (27.4%) compared to Lao Cai (8.7%)

### Seroprevalences of anti-hantavirus and anti-HEV antibodies

Of the 210 participants recruited from two provinces, serum samples were successfully obtained from 207. Overall, the AP of hantavirus IgG was 8.7% (95% CI: 5.4–13.6), while the TP was 4.7% (95% CrI: 0.2–11.1). Lao Cai exhibited higher AP and TP for hantavirus IgG compared to Dong Nai. Notably, the TP of hantavirus IgG was approximately half the AP in both the overall population and specifically in Dong Nai.

Given that the hantavirus IgG seroprevalence was higher than previous findings in Vietnam and regional studies, raising concerns about potential recent or ongoing infection, IgM testing was performed. The overall AP of hantavirus IgM was 1.9% (95% CI: 0.6–5.2), while the TP was 1.7% (95% CrI: 0.1–4.7). Of note, all positive cases of hantavirus IgM were identified among participants from Dong Nai.

Regarding HEV IgG, the overall AP was 26.7% (95% CI: 20.8–33.2), while the TP was 27.1% (95% CrI: 21.3–33.4). Lao Cai had a higher seroprevalence compared to Dong Nai, almost two times higher (Table 2). IgM testing was not done for HEV because the HEV IgG seroprevalence was similar to prior studies in Vietnam and other countries in the region.

Fig 3 illustrates the geographic distribution of 77 positive samples (18 hantavirus IgG, 4 hantavirus IgM, and 55 HEV IgG) among 70 wildlife farmers (33.8%). Of these, 21 (10.1%) were seropositive for hantavirus (IgG and/or IgM) and 55 (26.6%) were seropositive for HEV IgG. S2 Table summarizes the serological results by demographic characteristics and reported activities.

Notably, antibodies to more than one pathogen were observed in: (i) one case in Dong Nai that exhibited both hantavirus IgG and IgM antibodies; (ii) four cases (three in Lao Cai and one in Dong Nai) that had both hantavirus IgG and HEV IgG antibodies; and (iii) two cases in Dong Nai that were positive for both hantavirus IgM and HEV IgG antibodies. No individuals tested positive for all three types of antibodies.

### Self-reported symptoms of the most serious sickness during the last 12 months of serologically positive wildlife farmers

A total of 27 out of 70 (38.6%) serologically positive wildlife farmers reported having experienced illness or health issues in the last 12 months. The symptoms were variable, but the most common were fatigue (66.6%), headache (59.3%), cough (59.2%), fever (40.7%), and shortness of breath (22.2%) (Fig 4).

Among the 18 hantavirus IgG-positive participants, 8 (44.4%) reported experiencing symptoms, most commonly fatigue (n = 5), fever (n = 3), cough (n = 3), and headache (n = 2). Less frequently reported symptoms included shortness of breath, difficulty breathing, and sneezing/runny nose. Of the four hantavirus IgM-positive individuals, only one person reported

**Table 2. Apparent seroprevalence (AP) with 95% CI and true seroprevalence (TP) with 95% credible interval for hantavirus and hepatitis E virus in sera samples of wildlife farmers.**

| Province | Positive samples/ Tested samples | AP (%) with 95% CI | TP (%) with 95% credible interval |
|---|---|---|---|
| **Hantavirus (IgG)** | | | |
| Lao Cai | 5/46 | 10.9% (4.1–24.4) | 8.2% (0.4–20.9) |
| Dong Nai | 13/161 | 8.1% (4.5–13.7) | 4.6% (0.2–11.2) |
| Total | 18/207 | 8.7% (5.4–13.6) | 4.7% (0.2–11.1) |
| **Hantavirus (IgM)** | | | |
| Lao Cai | 0/46 | – | – |
| Dong Nai | 4/161 | 2.5% (0.8 - 6.6) | 2.2% (0.1–6.1) |
| Total | 4/207 | 1.9% (0.6–5.2) | 1.7% (0.1–4.7) |
| **HEV (IgG)** | | | |
| Lao Cai | 21/46 | 45.7% (31.2 - 60.8) | 46.7% (32.8–61.1) |
| Dong Nai | 34/161 | 21.1% (15.2 - 28.4) | 21.8% (15.7–28.6) |
| Total | 55/207 | 26.7% (20.8–33.2) | 27.1% (21.3–33.4) |

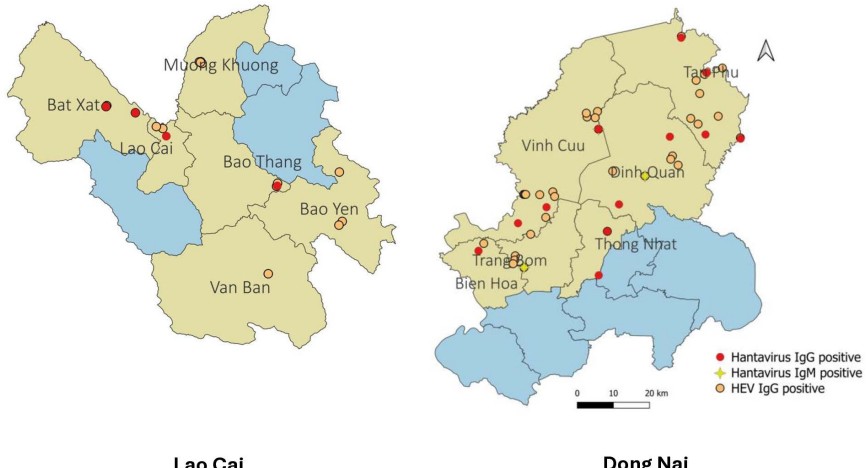

Lao Cai                Dong Nai

**Fig 3. Location of 77 serologically positive samples in study districts (yellow) in two provinces.** The map visually emphasizes the spatial distribution of these seropositive individuals, with markers representing their locations within these regions: red dots for hantavirus IgG, orange dots for HEV IgG, and yellow stars for hantavirus IgM (observed only in Dong Nai). Source of Vietnam administrative boundary data: https://data.humdata.org/dataset/cod-ab-vnm, accessed March 01, 2025, licensed under CC-BY-IGO.

symptoms consistent with acute hantavirus infection, including fever, fatigue, and cough, while the others were asymptomatic. Among the 55 HEV IgG-positive participants, 21 (38.2%) reported symptoms, including headache (n = 14), fatigue (n = 12), cough (n = 10), fever (n = 7), and shortness of breath (n = 5). Less commonly, participants reported sneezing or runny nose (n = 3), difficulty breathing (n = 3), vomiting (n = 2), diarrhoea (n = 2), muscle pain (n = 2), as well as weakness, rash, and hoarseness.

## Factors associated with seropositivity of hantavirus and HEV

Univariate analysis showed that wildlife farmers belonging to minority ethnic groups (OR = 3.2, 95% CI: 1.5–6.7), engaging in hunting or trapping wild animals (OR = 3.2, 95% CI: 1.1–10), consuming raw meat or wild animal

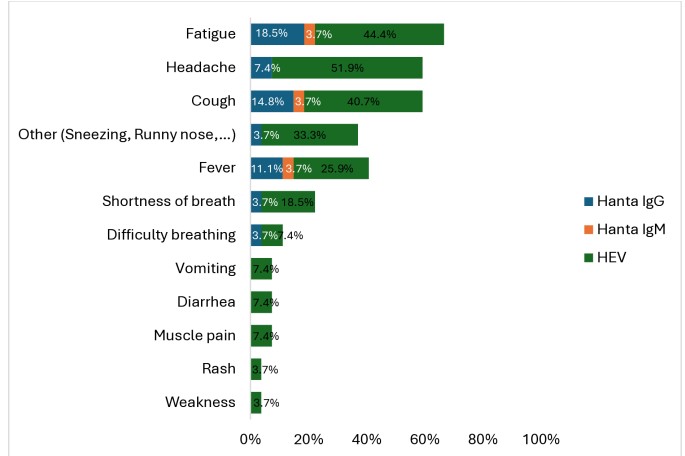

**Fig 4. Self-reported symptoms of serologically positive wildlife farmers of the most serious illness experienced within the past 12 months.**

products (OR = 5.8, 95% CI: 1.6–27), living in higher altitude (> 200 meters) (OR = 11.6, 95%CI: 2.9–77.2), and living in areas with higher mammalian diversity (>140 species) (OR = 2.5, 95%CI: 1.1–5.9) were statistically more likely to be seropositive against either hantavirus or HEV. Our analysis did not find a significant association between the species farmed and seropositivity for either hantavirus or HEV. Regarding anti-hantavirus seropositivity, farmers engaging in wildlife farming only (OR = 2.7, 95%CI: 1.1–6.9) was the only statistically significant association with infection.

Factors associated with anti-HEV seropositivity included identifying as a man (OR = 2.1; 95% CI: 1.1–4.2), belonging to an ethnic minority group (not Kinh) (OR = 4.6, 95% CI: 2.2–9.7), engaging in hunting or trapping wild animals (OR = 3.5, 95% CI: 1.2–10.5), and consuming raw meat or wild animal products (OR = 5.4, 95% CI: 1.6–21.3). Geographic variables were also significantly associated, including higher altitude > 200 meters (OR = 17.6, 95% CI: 4.4–117.4), and living in areas with greater mammalian diversity >140 (OR = 2.7, 95% CI: 1.1–6.4). (S3 Table in Supporting Information file).

Table 3 summarizes the results of a multivariable analysis exploring factors associated with seropositivity for anti-hantavirus and anti-HEV antibodies, as well as any seropositivity (combined seropositivity for either virus). Factors significantly associated with seropositivity for either hantavirus or HEV include men gender identity, consumption of raw meat or raw wildlife products, and living in high altitudes. Men had nearly double the odds of seropositivity compared to women (OR = 2.0, 95% CI: 1.0–3.9). Individuals who consumed raw meat or raw wildlife products were over five times more likely to be seropositive (OR = 5.6, 95% CI: 1.4–28.7). Furthermore, living at higher altitudes (>200 m) was strongly associated with seropositivity (OR = 13.6, 95% CI: 3.2–95.3), while those living at altitudes between 150–200m showed no significant association (OR = 1.2, 95% CI: 0.4–3.4). The Hosmer-Lemeshow goodness-of-fit (GOF) test for any seropositivity model showed $X^2 = 10.6$, df = 8, p-value = 0.2, indicating a good model fit.

The factor significantly associated with hantavirus seropositivity was engaging in farming wild animals only, with individuals involved in farming wild animals only having significantly higher odds of hantavirus seropositivity (OR = 2.7, 95% CI: 1.1–6.9). The GOF test for anti-hantavirus seropositivity model showed $X^2 = 11.2$, df = 8, p-value = 0.2, indicating a good model fit.

Factors significantly associated with HEV seropositivity were men gender, older age, consumption of raw meat or raw wildlife products, living in high altitudes and protective clothing use. Men were over three times more likely to be HEV

Table 3. Multivariable analysis of risk factors associated with anti-hantavirus and anti-HEV seropositivity.

| Factors | Any seropositivity | | Hantavirus | | Hepatitis E virus | |
|---|---|---|---|---|---|---|
| | OR (95%CI) | p-value | OR (95%CI) | p-value | OR (95%CI) | p-value |
| **Gender** | | | | | | |
| Man (vs woman) | 2.0 (1.0–3.9) | 0.05* | – | – | 3.1 (1.4–7.3) | 0.01* |
| **Age** | | | | | | |
| Older (vs younger) | 1.02 (1.0–1.1) | 0.06 | – | – | 1.03 (1.0–1.1) | 0.02* |
| **Consume raw meat or raw wildlife products** | | | | | | |
| Yes (vs no) | 5.6 (1.4–28.7) | 0.02* | – | – | 6.8 (1.6–31.8) | 0.01* |
| **Altitude** (vs group < 150 meters) | | | | | | |
| >200m | 13.6 (3.2–95.3) | 0.002* | – | – | 31.6 (5.5–204.4) | 0.0008* |
| 150-200m | 1.2 (0.4–3.4) | 0.74 | – | – | 1.4 (0.4–4.1) | 0.61 |
| **Engaging in farming wild animals only** | | | | | | |
| Yes (vs no) | – | – | 2.7(1.1–6.9) | 0.03* | – | – |
| **Wearing protective clothing when cleaning or contacting with wild animals** (vs never) | | | | | | |
| Always | – | – | – | – | 4.0 (1.4–11.2) | 0.008* |
| Sometimes | – | – | – | – | 0.9 (0.2–3.0) | 0.84 |

*Significant factors at p ≤ 0.05.

OR: odds ratio; CI: confidence interval.

seropositive compared to women, OR = 3.1, 95% CI: 1.4–7.3). For each additional year of age, the odds of HEV seropositivity increase slightly (OR = 1.03, 95% CI: 1.0–1.1). Farmers who reported consuming raw meat or wildlife products were over six times more likely to have HEV seropositivity (OR = 6.8, 95% CI: 1.6–31.8). Furthermore, consistent use of protective clothing when cleaning or contacting wild animals was significantly associated with higher odds of HEV seropositivity (OR = 4.0, 95% CI: 1.4–11.2), while occasional use showed no significant effect (OR = 0.9, 95% CI: 0.2–3.0). Farmers who lived at higher altitudes (>200 m) significantly increased the odds of HEV seropositivity (OR = 31.6, 95% CI: 5.5–204.4). The GOF test for anti-HEV seropositivity model showed $X^2 = 8.1$, df = 8, p-value = 0.4 implying that the data fitted well with the model.

## Discussion

This study explored the seroprevalence of antibodies to hantavirus and HEV in wildlife farmers in two provinces of Vietnam and identified potential risk factors for exposure. The AP among wildlife farmers in this study for hantavirus IgG was 8.7% with a TP of 4.7%, indicating past exposure to hantavirus among wildlife farmers. As outlined in the methods, AP and TP were calculated to account for test sensitivity and specificity of the diagnostic tests, which may bias estimates when the test performance is less than perfect.

Overall, the AP of IgG against hantavirus among wildlife farmers in this study was higher than global estimated (2.9%), and most continental figures [60], as well as previous studies in Vietnam (ranging from 1.6% to 3.7%) [30,49,50]. While also exceeding results from other countries [72–75], it was lower than reported seroprevalences in Cambodia (10%) [14], and Greece (9.7%) [76]. These differences can be explained by variations in handling of animals, cleaning the farming area, and potential exposure to contaminated environments, ecological factors, rodent population densities, farming practices, and human-animal interactions across contexts.

We further conducted hantavirus IgM tests to identify the recent infection with hantavirus to facilitate feedback to participants and local authorities. The seroprevalence for hantavirus IgM in our study was 1.9% (TP = 1.7%) which was

lower than another study conducted in the northern provinces of Vietnam (3.5%) [49]. However, it was higher than some studies conducted in other countries (Cambodia 0.2% [14], and Colombia 0.7% [73]). Notably, while Lao Cai had slightly higher seroprevalence (10.9% for IgG) compared to Dong Nai (8.1%), all IgM-positive cases were observed in Dong Nai, suggesting possible recent transmission events or ongoing exposure in this province. The absence of IgM-positive cases in Lao Cai may reflect either lower recent exposure or differences in surveillance timing. However, the number of IgM positive cases was too low to draw conclusions on why these cases emerged, and no hantavirus outbreaks have been officially reported in Dong Nai during the study period.

Overall, our anti-HEV IgG prevalence (26.7%) with TP = 27.1% was very similar to other studies in Vietnam [38,52]. Similar seroprevalence levels have been reported internationally, for example, 28.7% in the Netherlands [77], 18.4% in Cambodia [78], 21.5% in Thailand, 25.7% in Laos, and 33.5% in Myanmar [79]. While some specific subpopulations showed higher prevalence, such as forestry workers in France (31.0%) [80] and ethnic minorities in Ha Giang province (72.1%) [51], other studies reported lower rates (16.2% in Singapore [79], 11.9% in Indonesia, 11.3% in the Philippines [79], 12.4% in Ghana [81], 12.2% in Romania [82], 5.9% in Malaysia [79]). These variations reflected differing regional exposures, food practices, and environmental risk factors. Notably, our results showed that Lao Cai exhibited a higher HEV IgG prevalence (45.7%) compared to Dong Nai (21.1%), possibly linked to different consumption behaviours in each region; for example consumption of *tiet canh* – a traditional raw blood pudding made from fresh animal blood is more commonly consumed in northern Vietnam [83].

Our study revealed that only 38.6% of seropositive participants reported experiencing symptoms of illness within the past 12 months. It should be noted that antibodies, particularly IgG, can persist for years after infection. Therefore, they may have originated from an illness that occurred previously. The most common symptoms were fatigue, headache, cough, fever, and shortness of breath, which aligned with potential hantavirus and HEV infections. However, these reported symptoms may overlap with other infections (e.g., dengue, leptospirosis, typhoid) or non-infectious causes (e.g., occupational stress, environmental exposure), limiting the ability to attribute symptoms solely to hantavirus or HEV. Additionally, both hantavirus and HEV infections often present as mild or asymptomatic, particularly in healthy individuals [44,84]. Participants with mild symptoms may have forgotten past symptoms and might not have recognized them as significant or linked them to infection, especially if they were mild or resolved without medical intervention. This finding highlights the importance of raising awareness about zoonotic disease symptoms and encouraging individuals to seek medical care for any concerning symptoms, even if mild, to facilitate early detection and appropriate management of zoonotic diseases.

Our study identified several significant risk factors for seropositivity to hantavirus and HEV antibodies among wildlife farmers in Vietnam. Men exhibited higher odds of seropositivity for HEV compared to women. This finding could be explained by the gender role often observed in wildlife farming and related activities, where men are more likely to engage in tasks that involve direct contact with animals, such as hunting, slaughtering, or processing wildlife. This is consistent with the epidemiological findings of Dalton et al. [85], Aggarwal et al. [86] and Kamar et al. [32], that men were more likely to have been infected with HEV, with a man to woman ratio of around 3:1 [85]. However, more recent studies conducted by Berto et al. [38], Cao et al. [52] and Lichnaia et al. [51] found no significant association between gender and exposure to HEV. As such, whether gender plays a role will likely depend on local, cultural, and other contextual factors. Additionally, age showed a positive correlation with HEV seropositivity. This could indicate cumulative risk over time, as older individuals may have experienced prolonged exposure to wildlife or participated in risky practices for a longer period. This finding aligns with studies by Feng et al. [87], Berto et al. [38], and Mah et al. [55] which similarly showed that the HEV risk increases with age. However, it is also important to note the role of context specificity, as a study conducted by Cao et al. [52] recorded no significant association between age and HEV infection. Ethnicity was also reported as a significant factor, with non-Kinh (ethnic minority groups) exhibiting higher seropositivity. Vietnam has 54 officially recognized ethnic groups. The Kinh majority (87%) generally live in deltas and cities, while ethnic minorities (e.g., Tay, Thai, Muong, Hmong) (13%) inhabit mountainous and rural regions.

These groups may also have less access to health care services, public health information or preventive measures. This result was in line with other studies conducted in Ha Giang province [51] and in China [87].

Consumption of raw or undercooked wildlife products was strongly associated with HEV seropositivity, highlighting the role of dietary practices in faecal-oral transmission pathways. This finding was similar to previous studies and guidelines that consuming raw or undercooked meat poses the highest risk of hepatitis E infection [54,56]. In Vietnam, it is common that local people consume uncooked wild meat and wildlife products as they believe in the health benefits and exoticism of wild meat [88,89]. Therefore, it is suggested that health education programs on HEV prevention and control should be targeted at this high-risk population.

In addition, engaging in high-risk activities such as hunting or trapping wildlife was linked to increased odds of seropositivity for both hantavirus and HEV. Studies by Schielke et al. [90] and Baumann-Popczyk et al. [91] similarly found evidence of HEV infection among the hunters. Simpson et al. [92] and Bi et al. [93] likewise depicted that trappers, hunters, and farmers have a higher risk of contracting hantavirus infections. Our study also identified that only participating in wildlife farming was a specific risk factor for hantavirus exposure, underscoring the potential for pathogen spillover in farming environments. This finding suggests that direct contact with farmed animals poses a significant risk even without hunting, trading, or consuming wildlife. Several factors may contribute to this, including prolonged exposure to contaminated environments, and direct contact with infected animals and their waste such as animal blood, urine, and faeces, which are key transmission routes for both hantavirus and HEV. Additionally, poor farm biosecurity and limited use of protective equipment may further elevate exposure risks.

Geographic and environmental variables were significantly associated with seropositivity. Farmers living at higher altitudes (>200m) or in regions with greater mammalian biodiversity were more likely to be seropositive. Previous studies proposed that areas of higher biodiversity (e.g., the tropics) might confer a greater risk of zoonotic disease emergence under land-use change [94,95]. Nevertheless, other studies suggested that biodiversity loss may increase the likelihood of zoonotic disease emergence [96,97]. Our study did not directly look at biodiversity and so cannot contribute to this debate, but note that this is worth exploring further. Despite the observed dominant trend of increased disease emergence risk with higher mammalian richness, our finding neither definitively supports nor refutes the potential occurrence of a dilution effect for other zoonotic diseases. Regarding altitude, while direct studies correlating altitude with seroprevalence are limited, in rural and higher altitude areas in Vietnam, limited access to education, health care services, safe water, and proper sanitation could contribute to increased zoonotic disease exposure. Future research efforts should prioritize the underserved regions to investigate ecological, behavioural, and environmental factors influencing zoonotic transmission, enabling targeted interventions for infection prevention.

Interestingly, our findings indicated that consistent use of protective clothing was associated with higher odds of HEV seropositivity. However, this was based on self-reported use, and the state of clothing and their use was not assessed. A study conducted by Doos et al. [98] highlighted that regularly reusing protective clothing without adequate decontamination poses a significant risk of cross-contamination that can lead to the accumulation of pathogens. Moreover, improper doffing techniques can result in self-contamination. A study by Sahay et al. [99] indicated that errors in removing personal protective equipment (PPE), can increase the incidence of contamination. Guidelines also recommend always washing hands after removing protective clothing [100]. In addition, the PPE observed at the sample farms only included masks, gloves and protective clothing, relying solely on protective clothing without incorporating other PPE components, which may leave individuals vulnerable to infection. Research conducted in South Korea indicated that slaughter workers with HEV seropositivity reported higher use of protective vinyl gloves, aprons, boots, and disposable suits compared to other workers [101]. Similar findings were also observed in a study conducted by Bugeza et al. [102]. These suggest that either the equipment does not effectively prevent HEV infection or that it is not appropriately used. Nevertheless, Schielke et al. [90] provided evidence that wearing protective gloves during the disembowelling of wild boars significantly reduced the risk of HEV infection among hunters, highlighting that PPE can be effective when used

correctly in high-risk activities. Thus, we recommend that health information, education, and communication strategies using mass media and social media on the proper use of PPE should be organized for wildlife farmers to minimize the risk of infection.

Our finding of no significant association between farmed wildlife species and hantavirus or HEV seropositivity may reflect the shared exposures across farms in our study, such as environmental contamination or similar farming practices. Previous studies reported higher seroprevalence in individuals handling specific animals, such as rats for hantavirus [103] or wild boars for HEV [80]. Future studies with larger sample sizes for different species, incorporating molecular detection from animals and environmental samples, could help clarify the role of specific host species in viral transmission dynamics.

Our study found that several participants exhibited dual seropositivity for hantavirus IgG, IgM and HEV IgG, though no individuals tested positive for all three antibodies. The overlaps highlighted shared risk factors for these zoonotic pathogens, particularly high-risk behaviours and environmental exposures. The detection of HEV IgG in a greater proportion of participants compared to hantavirus antibodies can be explained by the differences in transmission dynamics. This finding underscored the need for education campaigns on zoonotic diseases, transmission pathways and hygiene practices.

This study had several possible limitations. First, the study was designed to capture a snapshot of the seroprevalence of hantavirus and HEV among wildlife farmers in Vietnam, as well as the risk factors influencing exposure at a single time point. We looked at who had been exposed to these diseases and who had not, but we could not assess how those, or potentially other, exposures directly led to infections. Future research would benefit from conducting longitudinal surveys or cohort studies to better link risk factors to disease outcomes. Second, this study asked about behavioural, wildlife activities and health issues as well as health-related symptoms over the past 12 months, which may be subject to recall bias or social desirability bias. Participants may have underreported high-risk behaviours, such as hygiene practices at the farm, consuming raw meat/raw products or engaging in hunting, leading to potential misclassification of exposure. Third, while the serological assays used in the study were robust, they have inherent variability in sensitivity and specificity. The anti-hantavirus IgG ELISA had sensitivity and specificity ranges of 75–98% and 91.4–100%, respectively, depending on the type of hantavirus. Moreover, the Wantai HEV IgG ELISA, while capable of detecting antibodies against the highly conserved ORF2 capsid protein of all HEV assays, does not distinguish among its strains, which include zoonotic (genotypes 3 and 4), rat HEV, and other genotypes of HEV. These may have led to the misclassification of serostatus, causing an incorrect risk factor analysis. It could be possible that the TP is subject to under or overestimation due to potential random errors associated with our measurement techniques. Further research employing genotype-specific or molecular assays can provide more information on typing the virus, thereby mitigating the risk of misinterpreting zoonotic prevalence and associated risk factors. Fourth, we used backward stepwise selection for multivariable modelling. While valuable for generating a parsimonious model in this exploratory study, this approach can increase risk for type 1 errors and exclusion of relevant but non-significant predictors [104]. To mitigate these concerns, we forced variables deemed biologically highly relevant, identified in the causal framework, to remain in the model. This strategy reflected the exploratory nature of our study and aligned with guidance that stepwise methods can be appropriate for early-stage investigations aimed at identifying potential predictors [105–107]. Future studies with larger samples and confirmatory designs should consider a full-model specification or Bayesian approaches to improve model stability and inference. Finally, the study focused exclusively on hantavirus and HEV. Other zoonotic pathogens that may pose risks to wildlife farmers were not assessed. This limitation reduced the ability to draw broader conclusions about zoonotic disease risks in this population. Despite these limitations, this study applied a One Health approach to examine farmer, animal, and environmental interactions, contributing a better understanding of zoonotic disease risks in wildlife farming. Future studies should expand the scope of pathogen investigation and consider employing a One Health approach to gain a more holistic understanding of zoonotic disease transmission dynamics.

## Conclusion

This study provided evidence that hantavirus and HEV were circulating among wildlife farmers in Vietnam, with high-risk groups including men, older individuals, ethnic minorities, and those engaged in risky practices such as wildlife farming, hunting, trapping, or consuming raw meat or products from wild animals that create risks like bat, bamboo rat, civet and wild boar. Geographic and environmental factors, including higher altitudes and biodiversity, also play a critical role in exposure risks. From a public health perspective, these findings underscore the urgent need for targeted interventions to mitigate zoonotic disease transmission in high-risk groups. Efforts should also address environmental management and promote safer farming practices to mitigate zoonotic disease risks in wildlife farming contexts. Mass media and social media platforms can serve as valuable tools for promoting zoonotic disease awareness. These platforms can be utilized to disseminate information regarding potential exposure risks associated with wildlife farming, educate the public on risk factors, and promote preventive measures including the proper use of PPE for effective disease control and prevention.

## Supporting information

**S1 File. Questionnaire.**
(PDF)

**S2 Table. Serological results among study participants by demographic characteristics and reported activities.**
(PDF)

**S3 Table. Univariable analysis of factors associated with any seropositivity, anti-hantavirus, and anti-HEV.**
(PDF)

## Acknowledgments

We would like to thank Dr. Bui Ngoc Anh, MSc. Nguyen Duy Quy and other staffs of the National Institute of Veterinary Research, National Institute of Animal Science, Department of Animal Health, Department of Forestry Protection and Centre for Disease Control in Lao Cai and Dong Nai provinces for their assistance in data and sample collection. We also thank Prof. Hasebe Futoshi, Dr. Abe Haruka and his team at the Vietnam Research Station of the Center for Infectious Disease Research in Asia and Africa, Nagasaki University Institute of Tropical Medicine within the premises of the National Institute of Hygiene and Epidemiology of Vietnam for their support in lab analysis.

## Author contributions

**Conceptualization:** Ha Thi Thanh Nguyen, Hu Suk Lee, Johanna F. Lindahl.

**Data curation:** Ha Thi Thanh Nguyen, Hu Suk Lee, Thang Nguyen-Tien, Sinh Dang-Xuan, Johanna F. Lindahl.

**Formal analysis:** Ha Thi Thanh Nguyen, Hu Suk Lee, Bernard Bett, Johanna F. Lindahl.

**Funding acquisition:** Bernard Bett, Hung Nguyen-Viet.

**Investigation:** Ha Thi Thanh Nguyen, Jiaxin Ling, Thang Nguyen-Tien, Sinh Dang-Xuan, Vuong Nghia Bui, Tung Duy Dao.

**Methodology:** Ha Thi Thanh Nguyen, Hu Suk Lee, Bernard Bett, Johanna F. Lindahl.

**Project administration:** Ha Thi Thanh Nguyen, Thang Nguyen-Tien, Sinh Dang-Xuan, Vuong Nghia Bui, Tung Duy Dao.

**Resources:** Ha Thi Thanh Nguyen, Hu Suk Lee, Bernard Bett, Jiaxin Ling, Hung Nguyen-Viet, Vuong Nghia Bui, Tung Duy Dao, Åke Lundkvist, Johanna F. Lindahl.

**Supervision:** Ha Thi Thanh Nguyen, Hu Suk Lee, Bernard Bett, Thang Nguyen-Tien, Sinh Dang-Xuan, Hung Nguyen-Viet, Vuong Nghia Bui, Tung Duy Dao, Johanna F. Lindahl.

**Validation:** Ha Thi Thanh Nguyen, Hu Suk Lee, Jiaxin Ling, Åke Lundkvist, Johanna F. Lindahl.

**Visualization:** Ha Thi Thanh Nguyen.

**Writing – original draft:** Ha Thi Thanh Nguyen.

**Writing – review & editing:** Ha Thi Thanh Nguyen, Hu Suk Lee, Bernard Bett, Jiaxin Ling, Thang Nguyen-Tien, Sinh Dang-Xuan, Hung Nguyen-Viet, Fred Unger, Steven Lâm, Vuong Nghia Bui, Tung Duy Dao, Åke Lundkvist, Genevieve Cattell, Johanna F. Lindahl.

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
