## [Decision Letter · Decision Letter 0]

5 Jun 2025

Dear Dr. Nguyen,

Thank you for submitting your manuscript to PLOS ONE. After careful consideration, we feel that it has merit but does not fully meet PLOS ONE’s publication criteria as it currently stands. Therefore, we invite you to submit a revised version of the manuscript that addresses the points raised during the review process.

We look forward to receiving your revised manuscript.

Kind regards,

Janin Nouhin, Ph.D.

Academic Editor

PLOS ONE

Journal Requirements:

3. Thank you for stating the following financial disclosure: [This work was part of the CGIAR initiative on One Health, which was supported by contributors to the CGIAR Trust Fund (https://www.cgiar.org/funders). In addition, JFL’s time was funded by FORMAS (2021–00833), HSL's time was supported by the National Research Foundation of Korea (NRF) grant funded by the Korea government (MSIT) (RS-2022-NR068754).]. 

4. We note that Figures 1 and 3 in your submission contain [map/satellite] images which may be copyrighted. All PLOS content is published under the Creative Commons Attribution License (CC BY 4.0), which means that the manuscript, images, and Supporting Information files will be freely available online, and any third party is permitted to access, download, copy, distribute, and use these materials in any way, even commercially, with proper attribution. For these reasons, we cannot publish previously copyrighted maps or satellite images created using proprietary data, such as Google software (Google Maps, Street View, and Earth). For more information, see our copyright guidelines: http://journals.plos.org/plosone/s/licenses-and-copyright.

1. You may seek permission from the original copyright holder of Figures 1 and 3 to publish the content specifically under the CC BY 4.0 license. 

Reviewers' comments:

Reviewer's Responses to Questions

**Comments to the Author**

1. Is the manuscript technically sound, and do the data support the conclusions?

Reviewer #1: Yes

Reviewer #2: Partly

2. Has the statistical analysis been performed appropriately and rigorously?

Reviewer #1: No

Reviewer #2: I Don't Know

3. Have the authors made all data underlying the findings in their manuscript fully available?

Reviewer #1: Yes

Reviewer #2: Yes

4. Is the manuscript presented in an intelligible fashion and written in standard English?

Reviewer #1: Yes

Reviewer #2: Yes

Reviewer #1: The authors report on a cross-sectional study of exposure to hantavirus and hepatitis E among farmworkers in two provinces in Vietnam. Along with reporting seroprevalence, they also report risk factors for exposure and found that gender, age, ethnicity, geography, and behaviors associated with greater wildlife interaction were associated with seropositivity. This study has the potential to make a useful contribution to the study of hantavirus and HEV within a One Health framework, but several key issues need to be addressed.

Major Comments

1. A major aim of the study is to assess risk factors for hantavirus and HEV, but the introduction contains minimal discussion of what we know about risk factors for these specific viruses from prior studies in other settings. The paper would be strengthened if this prior work were discussed in the introduction and used to motivate the key variables that the authors include in their analysis: age, gender, geography, and behaviors that lead people to interact more closely with wildlife. The authors provide a rich discussion of these risk factors in the discussion and could use some of these details as background and motivation for the analysis.

2. Backward selection can often lead to false positives (Type I errors) and can also fail to identify important predictors that might be excluded during the selection process. I suggest the authors consider fitting their model with all the predictor variables they think important for understanding infection patterns and base their inference on that complete model. This paper provides more detail on issues with stepwise model fitting:

Mundry, R., & Nunn, C. L. (2009). Stepwise model fitting and statistical inference: turning noise into signal pollution. The American Naturalist, 173(1), 119-123.

Minor Comments

Abstract

1. Line 26: Report the type of blood sample.

2. Line 36: Clarify the reference category for “wildlife farming activities only.”

Introduction

1. The last paragraph mentions the “distinct wildlife trade dynamics” of Lao Cai and Dong Nai. It would be helpful to have some description of these dynamics here in the Introduction. How do the two regions differ and how might these differences affect exposure?

Methods

1. What percent of farms in Dong Nai were recruited for the study? What were the characteristics of the farms selected for the study?

2. It is unclear how many participants were recruited from each farm. Were the samples relatively similar across farms? If not, how did they differ?

3. Did the authors consider another level of nestedness in their model for farm? It is unclear how many participants were recruited from each farm, but a three-level model with both farm and province as random effects might be appropriate.

Discussion:

1. Lines 450-452: Please include a reference for this statement (if possible): "In Vietnam, it is very common that local people consume uncooked wild meat and wildlife Products as they believe in the health benefits and exoticism of wild meat."

Reviewer #2: Nguyen et al. investigated antibody prevalence and risk factors of wildlife farmers (keeping wild boars, civets, bamboo rats or bats) for infections with hantavirus or hepatitis E virus (HEV) in Vietnam. About 200 wildlife farmers from 2 distant regions in Vietnam participated in the study. Hantavirus IgG was detected in 8.7 %, hantavirus IgM in 1.9% and HEV-specific IgG in 26.7% of tested persons. The only identified risk factor for hantavirus infection was wildlife farming activities, whereas for HEV infection men gender, older age, consumption of raw meat and residing in higher altitudes were identified as risk factors. Somewhat surprisingly, use of protective clothing was also identified as HEV risk factor. The authors conclude that improved biosecurity, proper hygiene practise and surveillance activities are necessary to prevent infections with hantavirus and HEV in future.

General Comments:

Investigation on risk factors for specific infections are of general importance for risk mitigation strategies to prevent disease in future. The two analyzed viruses – hantavirus and HEV – are important zoonotic antigens and wildlife farmers are generally considered to be of risk of these zoonotic infections. Although the results of the study are mainly of confirmational nature, the data are of general interest and might contribute to further develop specific prevention strategies. However, several aspects need to be addressed before publication of the manuscript.

First of all, the transmission pathways of HEV are insufficiently described, which might result in misinterpretation of findings. HEV can be grouped into four main human-pathogenic genotypes, of which only genotypes 3 and 4 are zoonotic and infect animals, especially pigs and wild boars. In contrast, genotypes 1 and 2 only infect humans and cannot be zoonotically transmitted. It is therefore misleading to include case numbers, which are derived from genotype 1 and 2 infection (see Ref. 26), directly before mentioning animal infections (lines 67-71), and mix all together. The situation is even more complicated as another different virus species, ratHEV, can infect rats and in some cases zoonotically transmitted to humans. Therefore:

- these genotypes/virus species, their transmission pathways and reservoirs has to be described in more detail

- the situation regarding the presence of these genotypes and ratHEV in Vietnam should be mentioned

- the specificity of the applied serologic test for detecting antibodies of the different HEV genotypes/virus species has to be included

- it would also be important to discuss the results in light of the genotypes/virus species host specificity, e.g. by comparing farmers keeping wild boar or bamboo rats vs. those keeping civets or bats.

The discussion is very long, also repeating already mentioned facts (e.g., in 1st paragraph) and should be condensed to the necessary information.

Additional specific comments:

- L. 26: It would be helpful to insert the animal species kept by the farmers (bats, bamboo rats, civets, wild boars) into the abstract.

- L. 40-41: An explanation on the finding with the protective clothing should be inserted as the conclusions (improved biosecurity, proper hygiene, …) are confusing as they stand now. Please insert “…although their proper use was not assessed” in l. 41, or similar.

- L. 65-71: see General Comments on HEV genotypes and species, their animal reservoirs, transmission pathways and occurrence in Vietnam.

- L. 187-191: see General Comments on specificity of the test for HEV genotypes and species.

- L. 282: read … (IgG and/or IgM) …

- L. 382-383: Has there been an increased number of illnesses in the Dong Nai province, which might indicate a hantavirus outbreak?

- L. 399-406: see General Comments on HEV genotypes/species and animal reservoirs.

- L. 407-408: it should be considered that antibodies can persist for more than one year. Therefore, they might originate from illness a longer time ago.

- L. 413-418: Did the hantavirus IgM-positive persons reported hantavirus-specific symptoms?

- L. 428: Please specify if “affected by HEV” relates to HEV infection or HEV-induced disease.

- L. 487-504: Please include in discussion the finding of Ref. 72, which described evidence for the benefit of protective gloves during disemboweling of wild boars, for preventing HEV infection.

- l. 513: see General Comments on HEV genotypes and species and their transmission pathways.

**Do you want your identity to be public for this peer review?** For information about this choice, including consent withdrawal, please see our Privacy Policy

Reviewer #1: No

Reviewer #2: No

---

## [Author Response · Author response to Decision Letter 1]

2 Jul 2025

We thank the reviewers and editors for their constructive feedback. A detailed, point-by-point Response to Reviewers and Editors document has been attached to this submission to address all comments raised.

Should you require any further clarifications or revisions, please do not hesitate to let us know.

---

## [Editor Report · Decision Letter 1]

18 Jul 2025

Seroprevalence and risk factors of hantavirus and hepatitis E virus exposure among wildlife farmers in Vietnam

PONE-D-25-15935R1

Dear Dr. Nguyen,

We’re pleased to inform you that your manuscript has been judged scientifically suitable for publication and will be formally accepted for publication once it meets all outstanding technical requirements.

Kind regards,

Janin Nouhin, Ph.D.

Academic Editor

PLOS ONE
---

## [Editor Report · Acceptance letter]

PONE-D-25-15935R1

PLOS ONE

Dear Dr. Nguyen,

I'm pleased to inform you that your manuscript has been deemed suitable for publication in PLOS ONE. Congratulations! Your manuscript is now being handed over to our production team.

Kind regards,

on behalf of

Dr. Janin Nouhin

Academic Editor

PLOS ONE